



# Monsoon-driven changes in aeolian and fluvial sediment input to the central Red Sea recorded throughout the last 200,000 years

Werner Ehrmann[1], Paul A. Wilson[2], Helge W. Arz[3], Hartmut Schulz[4], Gerhard Schmiedl[5]

[1]Institut für Geophysik und Geologie, Universität Leipzig, Leipzig, 04103, Germany
[2]University of Southampton, Waterfront Campus, National Oceanography Centre, Southampton, SO14 3ZH, United Kingdom
[3]Leibniz-Institut für Ostseeforschung, Warnemünde, 18119, Germany
[4]Fachbereich Geowissenschaften, Universität Tübingen, Tübingen, 72074, Germany
[5]Institut für Geologie, Centrum für Erdsystemforschung und Nachhaltigkeit, Universität Hamburg, Hamburg, 20146, Germany

*Correspondence to*: Werner Ehrmann (ehrmann@uni-leipzig.de)

**Abstract.** Climatic and associated hydrological changes controlled the transport processes and composition of the sediments in the central Red Sea during the last ca. 200 kyr. Three different source areas for mineral dust are identified. The dominant source is located in the eastern Sahara (Sudan and southernmost Egypt). We identify its imprint on Red Sea sediments by high smectite and Ti contents, high $^{87}Sr/^{86}Sr$ and low $\varepsilon_{Nd}$. The availability of deflatable sediments was controlled by the intensity of tropical rainfall and vegetation cover over northern Africa linked to the African monsoon. Intense dust input to the Red Sea occurred during arid phases, low input during humid phases. A second, less significant source indicated by palygorskite input is probably located on the eastern Arabian Peninsula and/or Mesopotamia, while the presence of kaolinite suggests an additional minor dust source in northern Egypt. Our grain size data reflect episodes of fluvial sediment discharge to the central Red Sea and document the variable strength in response to all of the precession-paced insolation maxima during our study interval including both those that were strong enough to trigger sapropel formation in the eastern Mediterranean Sea and those that were not. The African Humid Period most strongly expressed in our Red Sea record was the one during the Eemian last interglacial at ca. 125 ka, followed by those at 198 ka, 108 ka, 84 ka and 6 ka.



# 1 Introduction

North Africa and the Arabian Peninsula are characterised by strong latitudinal hydrological gradients (e.g., Jolly et al., 1998;
Gasse, 2000; Enzel et al., 2015) and feature the largest expanse of hot desert on Earth. The landscape is characterized by large
endorheic basins made up of ergs or sand seas such as the Great Sand Sea of the eastern Sahara and the Rub´ al Khali covering
much of the southern Arabian Peninsula, as well as desert pavements and desiccated alluvial and lacustrine deposits. These
regions are responsible for more than half the global atmospheric load of mineral dust (Tegen et al., 2002; Engelstaedter et al.,
2006; Stuut et al., 2009; Kumar and Abdullah, 2011; Schepanski et al., 2012; Schepanski, 2018; Kunkelova et al., 2022). Once
aloft, dust from North Africa and Arabia can be transported vast distances across the Atlantic and Indian Oceans, as well as
deposited more locally in the Mediterranean Sea, Red Sea and northern Arabian Sea (e.g., Chester et al., 1977; Sirocko and
Lange, 1991; Notaro et al., 2013; Scheuvens et al., 2013; Ramaswamy et al., 2017; Palchan and Torfstein, 2019; Barkley et
al., 2021). Properly read, archives of sea-floor sediments that contain windblown dust and riverine material provide a valuable
record of change in continental hydroclimate through geological time (e.g., Tiedemann et al., 1994; McGee et al., 2010;
Skonieczny et al., 2019). This approach shows that North Africa has a long history of radical astronomically driven change in
hydroclimate, shifting back and forth between arid and dusty conditions and humid vegetated ones since at least 11 million
years ago (Crocker et al., 2022).

Extensive analysis of the late Quaternary geological and archaeological records suggests that, first-order, the pronounced shifts
between arid and humid conditions was driven by amplification of seasonal northward migration of the Intertropical
Convergence Zone and the tropical rain belt in response to boreal summer insolation maxima driven by the precession of
Earth's axis. This resulted in phases of intensified African monsoon circulation and African Humid Periods (AHPs)
(Rossignol-Strick, 1983; Tjallingii et al., 2008; Parton et al., 2015; Grant et al., 2017). During the AHPs, more meteoric water
was available, and river systems, wetlands and lakes existed. The enhanced humidity was associated with a greening of the
Sahara and parts of the Arabian Peninsula (e.g., Larrasoaña et al., 2013; Nicholson et al., 2020; Drake et al., 2022). These
changes in North African hydrology also triggered sapropel formation in the eastern Mediterranean Sea (e.g., Rohling et al.,
2015) via increased freshwater flux down the Nile and the reactivation of ancient northward draining river systems in Libya
(Osborne et al., 2008; Blanchet et al., 2021). The AHPs also likely provided temporary pathways for modern human dispersal
(e.g., Rosenberg et al., 2011; Coulthard et al., 2013; Timmermann and Friedrich, 2016; Timmermann et al., 2022; Schaebitz
et al., 2021). The humid intervals were followed by dry conditions with scarce to no vegetation cover and strong dust transport,
comparable to present-day conditions. Proxy records and modelling results indicate that the last five AHPs and their subsequent
dust pulses differed in strength, duration and rate of change, in which the strongest AHP was associated with the last interglacial
maximum (Ehrmann et al., 2017; Ehrmann and Schmiedl, 2021; Duque-Villegas et al., 2022). Regional differences are
documented for the Holocene AHP, especially a W-E gradient with an earlier but more gradual termination of the AHP in the



east and a delayed but more abrupt termination in the west (Renssen et al., 2006; Brovkin et al., 2008; Shanahan et al., 2015; Dallmeyer et al., 2020).

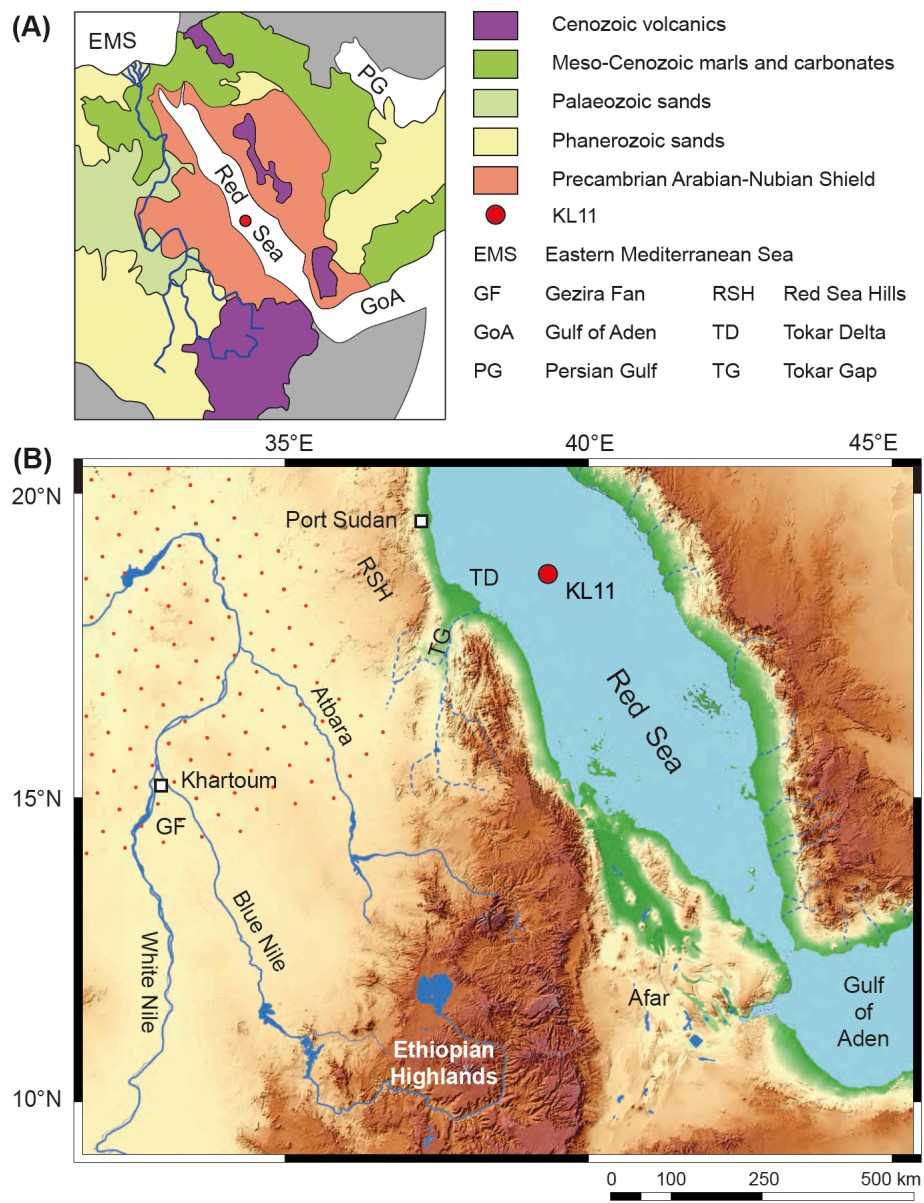

**Figure 1: Location of the investigated core KL11 in the Red Sea. (A) Overview of the Red Sea and the geology of its
hinterland (after Stein et al., 2007, and Palchan et al., 2013). (B) Topography of the area around the central and
southern Red Sea. The main dust source of the Eastern Saharan Province (ESP) is indicated by hachures (after
Kunkelova et al., 2022). Topographic map source: OpenTopoMap (CC-BY-SA).**



Reconstructions of the late Quaternary hydroclimate of North Africa are mainly built on records from marine sediment cores taken from the North Atlantic Ocean (e.g., deMenocal et al., 2000; Tjallingii et al., 2008; Scheuvens et al., 2013; Tierney et al., 2017) and the eastern Mediterranean Sea (e.g., Larrasoaña et al., 2003; Revel et al., 2010; Bout-Ramazeilles et al., 2013; Ehrmann et al., 2016, 2017; Blanchet et al., 2021; Ehrmann and Schmiedl, 2021). Fewer records are available from the Red Sea (Stein et al., 2007; Palchan et al., 2013; Palchan and Torfstein, 2019; Rojas et al., 2019; Hartman et al., 2020), but its position between the North African and the Arabian deserts makes it an important modern-day sink for mineral dust without the complicating influence of sediment input from large perennial rivers. These factors, together with its north-south orientation means that the Red Sea is well-suited to studying meridional change in past hydroclimate in a region where those changes likely exerted a strong control on evolutionary outcomes, including range expansions of our hominin ancestors (e.g., Groucutt et al., 2021; Scerri et al., 2021).

Here we present a sedimentary record from a marine core, KL11, taken from the central Red Sea, spanning the last ca. 210 kyr (Fig. 1). Our records provide a detailed reconstruction of aeolian and fluvial change because our study core was recovered seaward of the Tokar Gap (Fig. 1B), which is a major outlet for North African dust (Hickey and Goudie, 2007), and the Tokar Delta (Trommer et al., 2011), which owes its existence to localised fluvial sediment discharge from Africa during past humid periods. We combine high-resolution grain size, clay mineral and geochemical data, together with Nd and Sr isotope data to identify provenance and reconstruct changes in aridity/humidity through time.

## 2 Materials and methods

We investigated sediment core KL11, recovered from the central Red Sea at 18°44.5′ N and 39°20.6′ E from a water depth of 825 m during RV Meteor expedition M5/2 in 1987 (Fig. 1). The recovered section is 21.0 m long, and sediments comprise mainly terrigenous silt with varying amounts of biogenic carbonate, consisting of calcareous nannofossils, foraminifera and pteropods. The sediments have mostly a greyish, yellowish or brownish colour. No tephra layers were detected. The core does not show any indications of slumping, debris flows, or turbidites, but indicates simple undisturbed 'layer-cake' sedimentation. A lithified calcareous interval occurs between 90 and 193 cm depth.

### 2.1 Chronology

The age model for the upper 901 cm of KL11 was adopted from Hartman et al. (2020). They used the revised age model of Grant et al. (2012), which is based on a comparison of the planktic foraminiferal stable oxygen isotope record of KL11 with the U-Th dated speleothem record of Soreq cave, and integrated 8 AMS radiocarbon ages. The age model of the interval





between 901 and 1105 cm was established by comparison of the smectite record as a proxy for regional monsoon-driven hydrological changes (see below) with the U-Th dated composite Asian speleothem record (Cheng et al., 2016).

In Figs. 2–5, 7 and Suppl. Figs. 1–2 we indicate the timing of the sapropel layers in the eastern Mediterranean Sea that correspond to the AHPs. The ages of sapropels S1–S6 (S1: 5–10.5 ka; S3: 75–81.5 ka; S4: 97–105.5 ka; S5: 121–127 ka; S6: 165–174.5 ka) are taken from Ehrmann and Schmiedl (2021), the ages of S7 (191–197 ka) and S8 (213–220 ka) come from Emeis et al. (2003). The age models of cores from the eastern Mediterranean Sea are mainly based on a correlation of the $\delta^{18}O$

records to the LR04 stack (Liesicki and Raymo, 2005) and therefore may differ somewhat from the KL11 age model.

**2.2 Grain size, clay mineralogy and sediment geochemistry**

We sampled the upper 1105 cm of core KL11 at 2 cm spacing. The terrigenous components were isolated from bulk sediment by removing carbonate and organic matter with 10% acetic acid and 5% hydrogen peroxide, respectively. The terrigenous

matter content could be calculated by weight difference, because remains of siliceous microfossils were detected only sporadically in the sediments.

Grain size analysis of the terrigenous sediment fraction was performed using a laser particle sizer (Analysette 22, Fritsch GmbH) (Beuscher et al., 2020). We measured 155 size classes within the fraction 0–1000 µm. Concentrations of clay, fine silt,

medium silt, coarse silt and sand were calculated (Suppl. Fig. 1). The whole range of grain size data was fed into the opensource R-based endmember modelling programme RECA (Seidel and Hlawitschka, 2015) to identify similarities and variation patterns within the data set and to group them into subpopulations (endmembers, EMs). We used a model with 3 EMs which yields a high $r^2$ of 0.88 (mean coefficient of determination). Allowing 4 EMs yields only a minor increase in $r^2$ to 0.92.

The mineralogical composition of the clay fraction (<2 µm) was investigated by X-ray diffraction (XRD) following standard procedures (Ehrmann and Schmiedl, 2021). Major and trace elements were measured using an ITRAX (Cox Analytical Systems) XRF core scanner at the Leibniz-Institut für Ostseeforschung, Warnemünde, Germany. The scanner was equipped with a Cr-tube running at a tube voltage of 30 kV and a tube current of 55 mA. The exposure time was 5 ms per measurement, and the scanning interval was 0.2 cm. No data for calibrating and correcting for the water content were available. No XRF data

exist from the calcareous crust. Al, Si, K, Ti, Rb and Zr were assumed to describe the basic chemical composition of the terrigenous sediment fraction. These elements are reported normalised to total terrigenous element counts by calculating ratios of element counts / total terrigenous element counts (e.g., Ti/terr). Furthermore, the ratio of total terrigenous counts / Ca counts was calculated (Terr/Ca).



### 2.3 Radiogenic isotopes

All sample processing and analysis was undertaken at the University of Southampton's Waterfront Campus laboratories.

#### 2.3.1 Isolation of the terrigenous fraction of marine sediments

We treated our samples to remove all authigenic marine contaminants from the isotope fingerprint of the terrigenous fraction (carbonate, authigenic Fe-Mn oxyhydroxide coatings, organic carbon and marine barite) using the method of Jewell et al. (2022). That study demonstrated that small quantities of marine barite can have a large contaminating effect on the isotopic composition of the terrigenous fraction, especially for Sr. To remove marine barite, samples were leached with 0.2 M diethylene triamine pentaacetic acid (DTPA) in 2.5 M NaOH and left in a water bath set to 80 °C for 30 minutes. Following the recommended treatment protocol in Jewell et al. (2022), we repeated this step four times on three test samples to determine how many treatments were optimal for complete barite removal at this site. Marine sediments were considered free of marine barite when there was no appreciable decrease in Sr or Ba concentration, and no further change in measured $^{87}Sr/^{86}Sr$ of the silicate fraction with additional DTPA-NaOH treatment. Based on our test samples, we employed one DTPA-NaOH treatment (following the removal of all other marine phases).

#### 2.3.2 Analytical procedure

Our column chemistry procedure followed Jewell et al. (2022). Major element concentrations of sample residues were determined using a ThermoFisher iCAP6500 dual view inductively coupled plasma optical emission spectrometer (ICP-OES). Trace element concentrations were measured using quadrupole inductively coupled plasma mass spectrometer (ICP-MS Thermo Fisher Scientific XSeries 2). A suite of international rock standards was used for calibration (JB-1a, JGb-1, JB-2, JB-3, BHVO2, AGV-2, BCR-2, BIR-1), plus a spike of 5 ppb In, 5 ppb Re & 20 ppb Be as an internal standard. The precision for all measurements was <5.6 %.

#### 2.3.3 Sr isotopes

After column separation, the Sr fraction was dried down and loaded onto an outgassed tantalum filament with 1ml of a tantalum activator solution. The samples were analysed using a Thermo Fisher Scientific Triton Thermal Ionisation Mass Spectrometer using a multidynamic procedure and an $^{88}Sr$ beam of 2V. Fractionation was corrected using an exponential correction normalised to $^{86}Sr/^{88}Sr = 0.1194$. NIST987 was run as a standard in each turret and over the course of this study $^{87}Sr/^{86}Sr = 0.710241 \pm 0.000008$ (2sd; n = 20). The long-term average for NIST987 on this instrument is $0.710242 \pm 0.000021$ (2sd; n = 531).



### 2.3.4 Nd isotopes

Nd-isotope ratios were measured using a multi-collector ICP-MS (MC-ICP-MS, Thermo Fisher Scientific Neptune). Nd isotopic compositions were corrected following the method of Vance and Thirwall (2002) through adjustment to a $^{146}Nd/^{144}Nd$ ratio of 0.7219 and a secondary normalisation to $^{142}Nd/^{144}Nd = 1.141876$. For convenience the Nd isotope ratios are reported in epsilon notation ($\varepsilon_{Nd}$), where $^{143}Nd/^{144}Nd_{CHUR}$ represents the Chondrite Uniform Reservoir (CHUR) value of 0.512638 (Jacobsen and Wasserburg, 1980):

Equation 1:

$$\varepsilon_{Nd} = \left[ \frac{^{143}Nd/^{144}Nd_{sample}}{^{143}Nd/^{144}Nd_{CHUR}} - 1 \right] \times 10^4$$

### 3 Results

The results of our study are presented in Figs. 2–7 and Suppl. Figs. 1–2. Our sedimentary grain size data show three endmembers (EMs) with distinct modes (Fig. 2). EM1 has a dominant mode in the fine sand fraction at 65 µm and a subordinate shoulder at 20 µm. EM2 is clearly bimodal, with modes in the coarse silt fraction at 40 µm and 20 µm. EM3 is the finest endmember. It has a mode at 4.5 µm and a second mode at 16 µm. EM1 and EM2 show strongly fluctuating loadings downcore. Both have a distinct minimum around 125 ka, but otherwise they show opposing trends. The loadings of EM3 have a low background level, but distinct maxima occur at about 198, 125, 108, 84 and 6 ka.

The clay mineral assemblage of KL11 is dominated by smectite (Fig. 3). Maxima of about 40–50% occur around 208, 180, 152, 130, 114, 90, 63, 35 and 15 ka. Minima of about 25–30% occur around 196, 166, 142, 125, 108, 82, 52, 20 and 6 ka. Illite (15–32%), palygorskite (4–13%) and chlorite (10–20%) show generally similar distribution patterns, opposite to that of smectite. Kaolinite concentrations are relatively uniform at 15–20%, but manifest a distinct minimum at ca. 125 ka.

The Ti/terr ratio shows essentially the same temporal distribution pattern as the smectite concentration, while the K/terr ratio exhibits an inverse pattern to those two records (Fig. 4). Terr/Ca correlates well with the concentration of terrigenous matter, which ranges between 20% and 90% (Suppl. Fig. 2). Terr/Ca ratios show substantial millennial-scale variability. Elevated values occur during the glacial periods, lowest values at ca. 198, 125, 108, 84 and 6 ka.

The radiogenic isotope data on the terrigenous fraction in samples from KL11 show two main characteristics. First, both $\varepsilon_{Nd}$ and $^{87}Sr/^{86}Sr$ show clear coherent structure in their downcore variability (Fig. 5). Second, the neodymium isotope data fall in





a narrow range ($\varepsilon_{Nd}$~ -0.5 to -2.5) and the same is true of the $^{87}Sr/^{86}Sr$ isotope data (~0.7055 to 0.7075) except for one sample

taken from sediments of mid-Eemian age that is distinctly more radiogenic in composition (~0.7096).

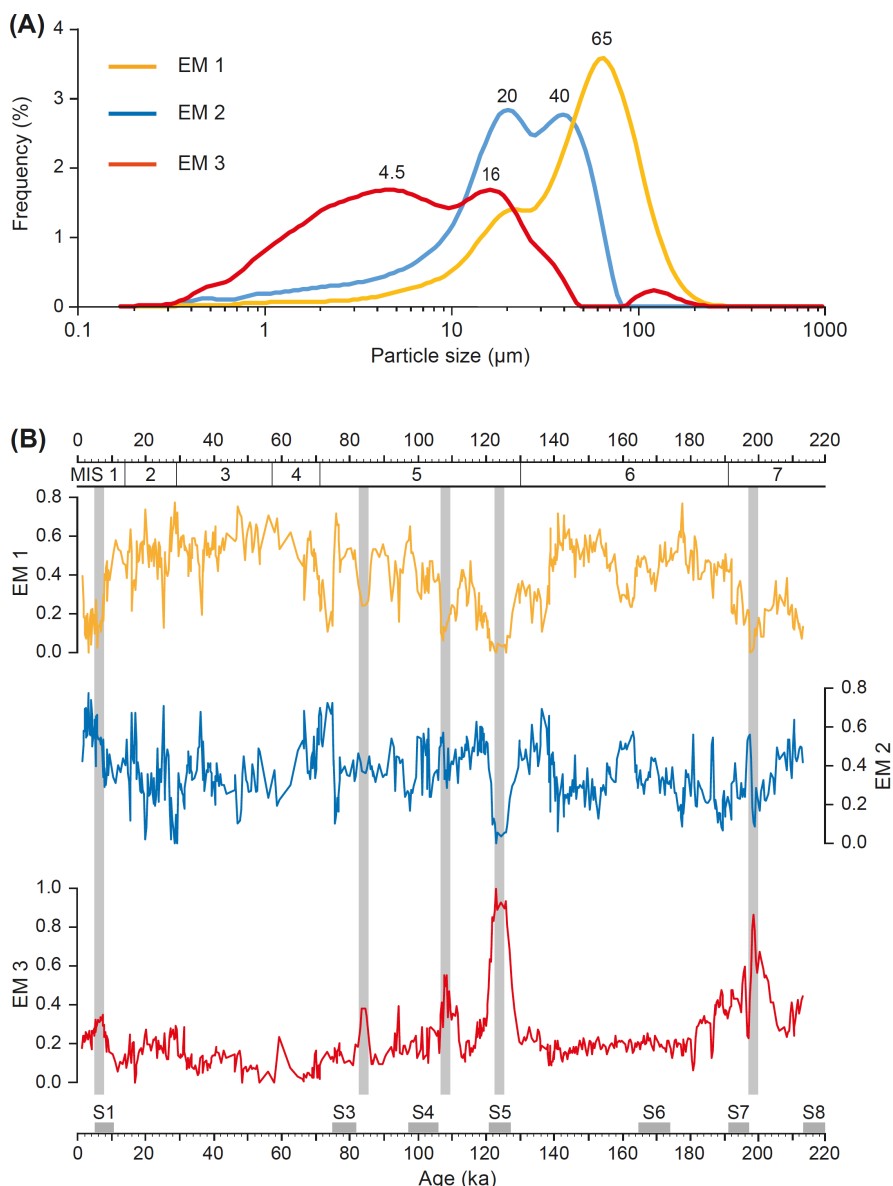

**Figure 2: Results of endmember modelling on the grain size distribution of the terrigenous sediment fraction, core KL11. (A) Grain size distributions of the three endmembers. The respective modes are given in μm. (B) Loading**

**patterns of the grain size endmembers. The main peak humid periods as inferred from EM3 are indicated by vertical grey bars. Marine isotope stages (MIS) are indicated at the top; horizontal bars at the bottom indicate sapropel layers S1–S8 in the eastern Mediterranean Sea associated with the African Humid Periods.**





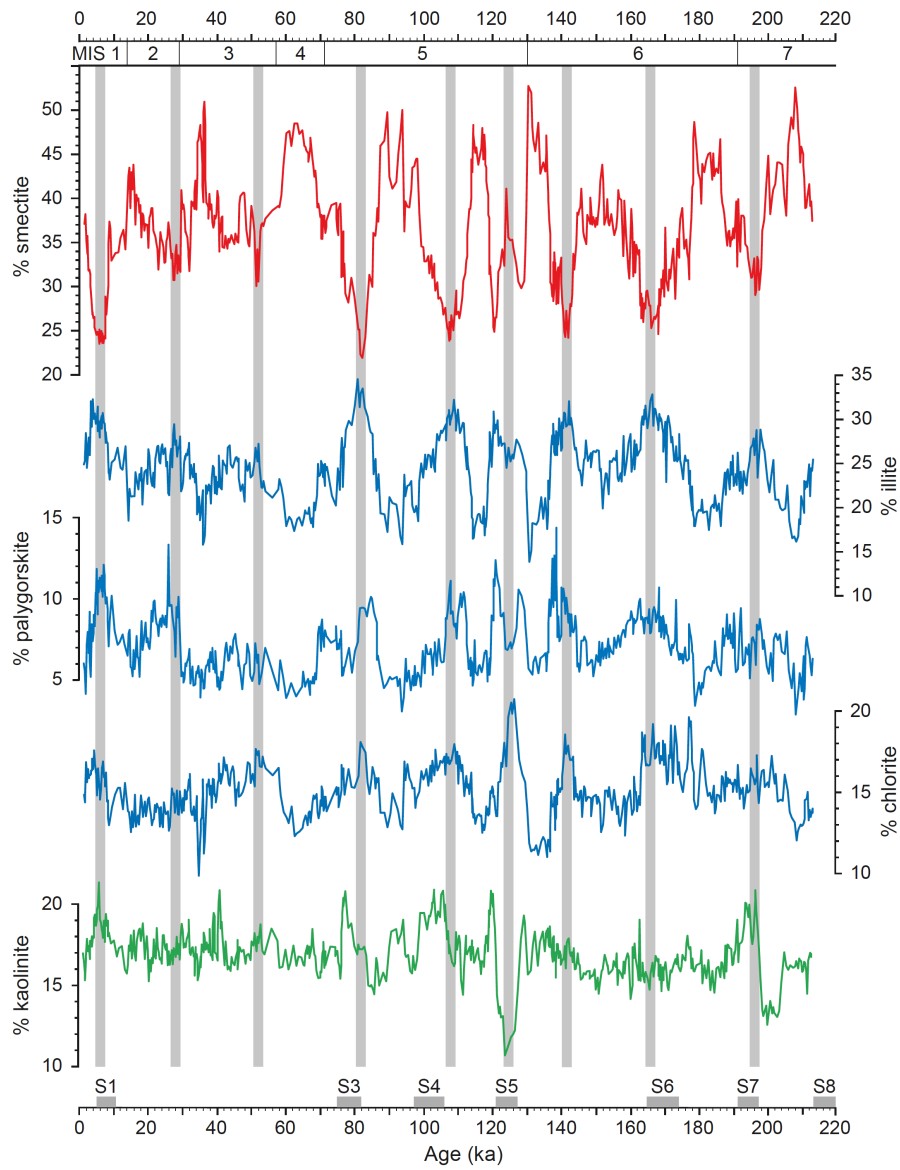

**Figure 3: Clay mineral composition of sediments from core KL11. The main peak humid periods as inferred from**
200 **minima in the smectite concentrations are indicated by vertical grey bars. Marine isotope stages (MIS) are indicated at**
**the top; horizontal bars at the bottom indicate sapropel layers S1–S8 in the eastern Mediterranean Sea associated with**
**the African Humid Periods.**



## 4 Discussion

### 4.1 Evaluation of proxies for aeolian and fluvial sediment transport to the central Red Sea

The absence of active perennial rivers entering the central Red Sea means that aeolian transport is the main mechanism of sediment delivery to the basin today. Some fluvial sediment discharge is suggested to have occurred during past humid phases, especially during the Eemian (Arz et al., 2003; Palchan et al., 2013; Hartman et al., 2020). However, even then, riverine input was likely of minor significance overall because the wadis that run towards the central Red Sea drain only small catchment areas (Fig. 1B).

Both the concentration of terrigenous matter and the Terr/Ca ratios show substantial variability on orbital time scales. Elevated values generally occur during glacial periods (Fig. 4, Suppl. Fig. 2), distinct minima during the intervals that correspond to the sapropel horizons in the eastern Mediterranean at ca. 198, 125, 108, 84 and 6 ka. However, interpretations of these time series are subject to closed sum issues because they are affected by fluctuating concentrations of biogenic carbonate as well as changes in terrigenous input. Negative correlations to the abundance of planktic foraminifera and pteropods per gram dry sediment (Almogi-Labin et al., 1998; Suppl. Fig. 2) point to elevated carbonate production during interglacial sea-level high stands and concomitant periods of less extreme saline surface waters in central Red Sea. This implies that carbonate production is a major control of sediment composition. Similar issues complicate the hematite content deduced from magnetic data and the Ti/Ca records that both have been used to quantify the dust accumulation in core KL09, recovered ca. 185 km NW of KL11 (Roberts et al., 2011), and showing a similar pattern as our data from KL11.

Detrital grain size (Fig. 2) provides a way to distinguish between aeolian and fluvial transport processes based on the typical modes of the EMs (Beuscher et al., 2017, 2020; Rojas et al., 2019; Blanchet et al., 2021). A mode of 65 µm (EM1) is indicative of coarse dust transported over short distances or by high wind strength. EM2 has modes of 40 µm and 20 µm, in the loess grain size fraction, and is interpreted as fine-grained dust transported over larger distances or by weaker winds. The interpretation of EM1 and EM2 as dust is supported by the angularity of the coarse silt and sand grains visible under the microscope. We interpret EM3, with a main grain size mode of 4.5 µm, to represent riverine suspension. This endmember also shows a subordinate mode of 16 µm, well in the medium silt grain size range. The distribution of the medium silt displays a rough similarity with the well correlated fluvial fine silt and clay record (Suppl. Fig. 1). This may indicate a secondary, more local source for in the riverine sediment input. However, a similar mode of 20 µm is present in the aeolian EMs 2 and 3. We therefore infer that fluvial sediment influx was accompanied by a persistent influx of medium silt sized dust that is responsible, in part, for the 16 µm mode in EM3.

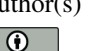



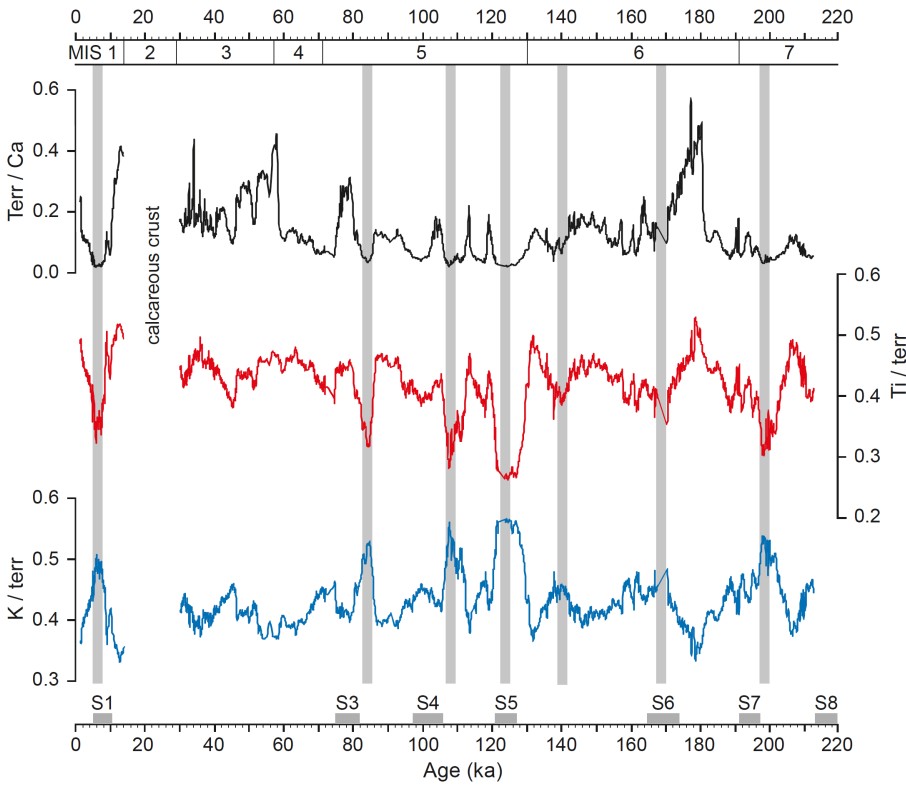

**Figure 4: XRF data from core KL11 (5-point running average). Terr/Ca gives the ratio between the sum of terrigenous elements and Ca. Ti and K were normalised by calculating their portion on the terrigenous elements. No data are available from the calcareous crust. The main peak humid periods as inferred from the XRF data are indicated by vertical grey bars. Marine isotope stages (MIS) are indicated at the top; horizontal bars at the bottom indicate sapropel layers S1–S8 in the eastern Mediterranean Sea associated with the African Humid Periods.**

The loadings of EM3 have a low background level, indicating that river discharge to the central Red Sea was a subordinate process through most of our study interval. However, EM3 displays distinct maxima during the monsoonal index maxima associated with AHPs 7, 5, 4 and 3 (Figs. 2, 7e) documenting intervals of local riverine sediment input.

Advection of coarse and fine dust was the dominant process of sediment transport to the central Red Sea during most of the investigated time interval, as indicated by the loadings of the EMs (Fig. 2). The coarse dust EM1 generally dominated during the glacial intervals and probably indicates stronger wind regimes. Minima occur during the humid phases. The influx of coarse dust ceased during AHP7 and AHP5, when fluvial influx was most intense. With the exception of the AHP5 minimum, EM2 loadings show a distribution opposite to EM1.




### 4.2 Sediment provenance and transport

The different patterns in the temporal distribution of the individual clay minerals, terrigenous elemental ratios and Nd and Sr isotope data of the KL11 sediments (Figs. 3–5) indicate that several independent source areas contributed to the aeolian and fluvial sediment input to the central Red Sea.

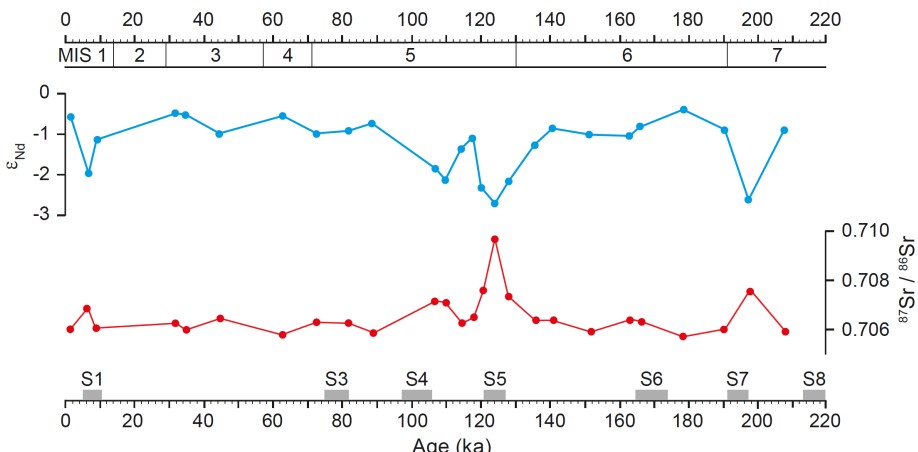

**Figure 5: Strontium and neodymium isotope records of the bulk terrigenous fraction in core KL11. Analytical**
**uncertainty on the isotope data is smaller than the size of the data points (see section 2.3). Marine isotope stages (MIS)**
**are indicated at the top; horizontal bars at the bottom indicate sapropel layers S1–S8 in the eastern Mediterranean Sea**
**associated with the African Humid Periods.**

Sources of mineral dust emission surround the Red Sea and are especially intense and proximal to the central Red Sea in the
eastern Sahara (Fig. 6A; Kunkelova et al., 2022), including immediately west of Tokar Gap, an approximately 110 km-wide lowland pass in the N-S trending Red Sea Hills, situated at ca. 18°N, some 50 km inland from the Red Sea coast (Fig. 1B). Dust transport through Tokar Gap is especially pronounced during summer (Jiang et al., 2009; Langodan et al., 2014; Kalenderski and Stenchikov, 2016). Wind takes up dust in southernmost Egypt and Sudan (Prospero et al., 2002; Schepanski et al., 2009; Bakker et al., 2019). This area is known as "Potential Source Area 6" (PSA 6; Scheuvens et al., 2013), or "Eastern
North African PSA" (Jewell et al., 2021), or "Eastern Sahara PSA" (Kunkelova et al., 2022), hereafter referred to as ESP. Dust is preferentially emitted from desiccated alluvial fans and plains, river beds, lake beds and wetlands (Tegen et al., 2002; Schepanski et al., 2009), such as Gezira alluvial fan south of Khartoum. This fan is characterised by presently inactive distributary channels of the Blue Nile and endorheic drainage systems (Williams, 2020).

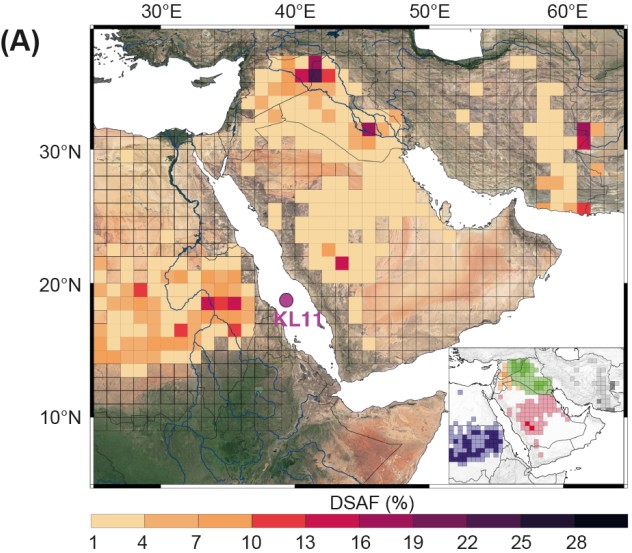

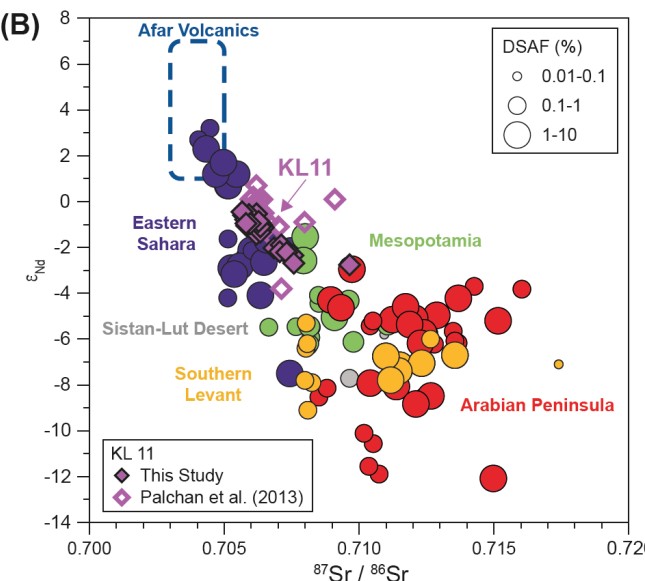


**Figure 6: Fingerprinting the provenance and transport pathway of the terrigenous fraction supplied to the central Red Sea (core KL11) using its Sr and Nd isotope composition. (A) Dust source activation frequency map of Northeast Africa and Southwest Asia (DSAF, calculated as the percentage of days within a given month wherein one or more dust events are recorded in a 1° x 1° grid cell; after Kunkelova et al., 2022, and references therein). Background: © Google Earth.**

**Inset shows the preferential dust source areas (PSAs) defined by Kunkelova et al. (2022) colour coded as in panel B. (B) Comparison of Sr and Nd isotope data from core KL11 (pink diamonds) compared to the composition of dust sources (circles). Circle size corresponds to DSAF (panel A) sampled. Note the strong correspondence between the radiogenic isotope composition of samples from KL11 and the geochemical fingerprint of the eastern Sahara.**



Sediments in the ESP are dominated by weathering products that are derived from the Cenozoic basalts in the Ethiopian Highlands and fluvially transported by the Blue Nile (Williams, 2020; Fig. 1). Their isotopic composition ($\varepsilon_{Nd}$ ~ -1.1, $^{87}Sr/^{86}Sr$ ~ 0.7059; Kunkelova et al., 2022; Fig. 6) shows the typical signature of Blue Nile sediments (Palchan et al., 2013). Blue Nile sediments are also characterised by high smectite concentrations derived from weathering of the basaltic material (Hamann et al., 2009; Revel et al., 2014; Ehrmann et al., 2016) and by high Ti contents (Revel et al., 2010; Hennekam et al., 2015). Thus,

sediments from the ESP can be identified in the central Red Sea by their radiogenic $\varepsilon_{Nd}$ and non-radiogenic $^{87}Sr/^{86}Sr$ signatures, high smectite contents and high Ti/terr ratios (Figs. 3–6). Some direct wind transport from the Ethiopian Highlands during glacial periods cannot be excluded, though the highlands are not dust-active today (Schepanski, 2018; Kunkelova et al., 2022).

Our Nd and Sr isotope data on the bulk terrigenous fraction from KL11 are so closely aligned with the composition of the ESP

that they can be explained by dust from this source alone (Fig. 6B). However, while the high smectite contents in these sediments suggest that the ESP is the main source, other clay minerals present also suggest there are contributions from secondary sources. Illite, chlorite and palygorskite show downcore variability opposite in sign to smectite and Ti/terr (Figs. 3, 4), and point to an additional source area, probably dominated by granitoid and metamorphic rocks and their weathering products. Palygorskite and illite are especially well correlated in our data. Palygorskite is the most diagnostic of these clay

minerals. It is a typical wind-blown mineral in sediments of the eastern Mediterranean Sea and a good tracer of Saharan dust. Sources for the eastern Mediterranean palygorskite are Paleogene sedimentary rocks and their weathering products located mainly from Tunisia to Egypt (Foucault and Meliérès, 2000; Goudie and Middleton, 2001; Bout-Roumazeilles et al., 2007; Scheuvens et al., 2013). However, northern Africa is not a likely source for the palygorskite in our records, because the aeolian sediments derived from there show a strong negative correlation between palygorskite and illite (Ehrmann and Schmiedl,

2021). Instead we record a positive correlation between these minerals in our downcore record from KL11. The ESP in Sudan and the Bodélé depression in Chad can be excluded as sources, because, so far, no palygorskite has been reported in dust originating from these sources (Bout-Roumazeilles et al., 2007; Scheuvens et al., 2013).

The Arabian Peninsula is a well-known source for wind-blown palygorskite (Sirocko and Lange, 1991; Sirocko et al., 1991).

The Upper Cretaceous and Tertiary sedimentary rocks, their weathering products and the soils in eastern and central Saudi Arabia are rich in palygorskite (Aba-Husayn and Sayegh, 1977; Mackenzie et al., 1984; Shadfan et al., 1984; Shadfan and Mashhady, 1985). Palygorskite is also widespread in the flood plains of Mesopotamia (Aqrawi, 1993; Al-Bassam, 2019). Both the lowlands of the eastern Arabian Peninsula and Mesopotamia are major dust emission areas (Hamidi et al., 2013; Jish Prakash et al., 2015; Ramaswamy et al., 2017; Kunkelova et al., 2022) and sources of dust to the Arabian Sea and the Red Sea

(Sirocko and Lange, 1991; Notaro et al., 2013; Jish Prakash et al., 2015; Ramaswamy et al., 2017). Our radiogenic isotope data from KL11 are almost entirely distinct from the relatively well-documented (Kunkelova et al., 2022) dust sources of the Arabian Peninsula, the single exception being a sample of Eemian age (Fig. 6B). The isotopic composition of Mesopotamian



dust is poorly documented (Kunkelova et al., 2022) but the data available suggest that windblown dust transport from this source may be responsible for the palygorskite deposited at KL11 (Fig. 6). A fluvial origin of palygorskite is unlikely, because there are no source and no potential sedimentary source rocks known for this mineral in the small mountainous catchments of the wadis entering the central Red Sea (U.S.G.S., 1963; G.M.R.D., 1981).

Kaolinite has a distribution pattern that differs from those of the other clay minerals requiring for a separate source. We infer that this clay mineral is derived from Egypt. High kaolinite concentrations have been reported there in the "Sinai Assemblage" and the "Egyptian Wadi Assemblage"; they originate mainly from Cretaceous and Cenozoic sedimentary rocks (Hamann et al., 2009). Transport is by the NNW winds blowing throughout the year along the axis of the Red Sea to site KL11 (Langodan et al., 2014; Ramaswamy et al., 2017).

Kaolinite is also reported from soils in the western Arabian Peninsula (Aba-Husayn et al., 1980; Shadfan et al., 1984). Derived from such a source, its temporal distribution pattern should be similar to that of palygorskite and illite, because the same transport processes would have been responsible for their supply. This, however, is not the case, as kaolinite shows an independent distribution pattern (Fig. 3).

It is likely that illite and chlorite contribute to the fluvial sediment fraction deposited during the humid phases. Illite and chlorite are typical clay minerals derived from physical weathering of granitoid and metamorphic rocks, as they are widespread in the Arabian-Nubian Shield and uplifted terrains on both sides of the central Red Sea (Fig. 1A). The most obvious route for fluvial sediment transport to the central Red Sea is via the Tokar Delta, which is fed by the seasonal Baraka River and its tributaries Anseba River and Langeb River (Fig. 1B). The Baraka catchment covers some 66.000 km². It is active for 40-70 days per year, mainly during autumn, with an annual water discharge of 200–970 x 10⁶ m³ (Trommer et al., 2011). The headwaters in the Red Sea Hills and Eritrean Highland (Hickey and Goudie, 2007) are composed of rocks of the Arabian Nubian Shield, mainly Precambrian gneisses, schists and granitoids (G.M.R.D., 1981). Only small wadis are found on the Arabian side of the central Red Sea. They reach <120 km into the peninsula and mainly drain rocks of the Arabian Nubian Shield and the coastal plain (U.S.G.S., 1963), and they are not likely major sources of fluvial sediment influx.

### 4.3 Climatic forcing of aeolian and fluvial sediment input

Our downcore records are characterised by strong cyclic changes in sediment composition (Figs. 2–5). They document powerful hydrological changes in the hinterland that were caused mainly by precession-driven changes in insolation, the intensity of the African monsoon and the S-N movement of the tropical rain belt (Fig. 7).







**Figure 7: Dry and humid phases of the last ca. 200 ka. (a) Monsoon index (W/m$^2$) relative to present, calculated after Rossignol-Strick (1983) by using the June insolation at 23.45°N and at the equator (Laskar et al., 2004); (b) Asian monsoon (U-Th dated composite δ$^{18}$O Asian speleothem record; Cheng et al., 2014); (c) Nile discharge as documented by the smectite/(illite + chlorite) ratio in core SL110 off Israel (Ehrmann et al., 2016); (d) monsoonal humidity in North**

**Africa as exemplified by Ti/Al ratios in sediments of ODP Site 968, Eratostenes Seamount, eastern Mediterranean Sea (Konijmendijk et al., 2014); (e) fluvial influx to the central Red Sea (KL11, EM3); (f) dust influx from the Eastern Sahara Province to the central Red Sea as documented by Ti/terr in KL11; (g) dust influx from the Eastern Sahara Province to the central Red Sea as documented by the concentration of smectite in KL11. Marine isotope stages (MIS) are indicated at the top; horizontal bars at the bottom indicate sapropel layers S1–S8 in the eastern Mediterranean Sea**

**associated with the African Humid Periods.**

The smectite concentrations and the Ti/terr ratios reflect hydrological changes that controlled the availability of mineral dust in the main source region for the KL11 sediments, the ESP. During humid periods, increased summer rainfall and longer wet seasons led to increased chemical weathering of the volcanic rocks in the Ethiopian Highlands, formation of smectite and high

sediment loads down the Blue Nile. The Nile load was partly deposited in the lowlands at the foot of the Ethiopian Highlands, in the ESP, and transported in suspension to the Nile Delta (Revel et al., 2010, 2014) and the eastern Mediterranean Sea (Fig. 7c). Hence, smectite maxima occur in sediments deposited beneath the Nile discharge plume off Israel during the AHPs (Ehrmann et al., 2016; Ehrmann and Schmiedl, 2021). Dust emission and aeolian sediment transport from the ESP to the central Red Sea was limited during these intervals as marked by minima in smectite concentrations, Ti/terr ratios and ε$_{Nd}$

values, and by maxima in $^{87}$Sr/$^{86}$Sr (Figs. 5, 7f, g). The transport of coarse dust decreased markedly during the AHPs, and almost ceased during AHPs 7, 5 and 1, probably due to less strong winds. A severe reduction of the fine-grained dust occurred only during AHP5 (Fig. 2). For AHP1, Palchan and Torfstein (2019) estimated a >50% reduction in eastward dust fluxes.

During dry periods, chemical weathering in the Ethiopian Highlands decreased and less sediment was transported down the

Nile (Fig. 7c). Vegetation cover was reduced and the sediments in the endorheic drainage basins and alluvial plains in the foreland of the Ethiopian Highlands desiccated and became prone to deflation by wind. Dust transport from the ESP via Tokar Gap to the central Red Sea became active. Thus, smectite maxima, enhanced Ti/terr ratios, a more radiogenic Nd and more non-radiogenic $^{87}$Sr/$^{86}$Sr compositions are observed in KL11 during dry phases and document active windblown transport. At the same time, dust transport from North Africa to the eastern Mediterranean Sea was enhanced as displayed by the clay

mineral record (Ehrmann and Schmiedl, 2021) and by XRF-based elemental ratios, especially Ti/Al, which were also used as a monsoonal humidity index (Fig. 7d; Konijnendijk et al., 2014).

The smectite concentrations of marine sediments in the central Red Sea are especially sensitive recorders of monsoon activity (Fig. 7g). Smectite minima align to the distinct AHPs 1 and 3–7 that were linked to strong insolation maxima and the intense





monsoon rains that led to enhanced humidity in North Africa (Fig. 7d) and to sapropel formation in the eastern Mediterranean Sea. They also align to the weaker insolation maxima and less intense humidity during MIS 2, 3 and 6b that did not lead to sapropel formation.

Both the smectite concentrations and Ti/terr record changes in dust influx from the ESP. The two records show the same main
pattern, but differ in detail (Fig. 7f, g). Ti/terr minima are somewhat shorter and sharper than smectite minima during humid periods. This may be due to the heterogeneity of the source. Smectite was measured in the clay fraction, Ti/terr in the bulk sediment. We infer that, during the peak dry phases, both coarse and fine particles were blown out from all areas of the ESP. When the climate became more humid, the coarser sediments of the channels, wadis and alluvial fans of the ESP were wetted by surface water and groundwater. They possibly became vegetated and ceased as dust sources. In contrast, the alluvial plains
still were dry and provided smectite-rich dust, until the peak humid phase was reached. This interpretation is consistent with the fact that the maxima in the fluvial EM3 and the minima in Ti/terr correlate well with each other. The peaks in these records correlating to AHP3 and AHP7 occur some 2 kyr earlier than the corresponding smectite minima (Fig. 7f, g) presumably signalling desiccation of the alluvial plains before their associated fans, channels and wadis.

The presence of palygorskite (Fig. 3) throughout our record, including in the humid periods, suggests persistent wind transport from Mesopotamia and possibly the Arabian Peninsula (but see 4.2) to the central Red Sea. According to records from lake sediments, drainage systems, speleothems and archaeological sites (Armitage et al., 2011; Rosenberg et al., 2011; Drake et al., 2013; Nicholson et al., 2020; Groucutt et al., 2021), and results from climate models (e.g., Dallmeyer et al., 2020, 2021), the Arabian Peninsula was significantly wetter during the humid phases, especially during the Eemian. The rainfall, however, was
confined to the summer months. Vegetation expanded particularly in the southern Arabian Peninsula, but desert conditions continued to exist in the northern and northeastern parts of the Arabian Peninsula (Jennings et al., 2015; Dallmeyer et al., 2020, 2021). In this interpretation, dust uptake was still possible in southern Mesopotamia and southern Arabia, mainly during the dry winter months. However, the fluctuating concentrations of the palygorskite do not allow us to reconstruct the intensity of dust transport, because they are influenced by dilution with smectite.


The kaolinite input (Fig. 3), inferred to be sourced from Egypt, is ascribed to northwesterly winds. The concentrations are relatively constant through time. However, a distinct minimum is documented for the pronounced AHP5. It is possible that a southward shift of the Mediterranean winter precipitation system (Kutzbach et al., 2020; Cheddadi et al., 2021) contributed to humidity in the northern Saharan and the Mediterranean regions during AHP5 which would have led to increased vegetation
cover and hampered dust uptake.

High loadings of the grain size EM3 (Fig. 7e) suggest fluvial sediment discharge reached KL11 during the AHPs. The maximum of AHP5 is most pronounced, followed by those of AHP7, AHP4, AHP3 and AHP1. The strength of the fluvial



sediment discharge signal to the central Red Sea therefore verifies the relative intensities of the individual AHPs as postulated

based on records of post-AHP dust pulses to the eastern Mediterranean Sea (Ehrmann et al., 2017; Ehrmann and Schmiedl, 2021) and in the modelling results of Duque-Villegas et al. (2022).

The Eemian (AHP5) maximum in the loadings of the fluvial grain size EM3 correlates well with a secondary maximum in the concentration of smectite, which otherwise is interpreted to be of aeolian origin and derived from the ESP (Figs. 2, 3, 7e, 7g).

This short and small smectite maximum, however, is probably of fluvial origin. The smectite cannot be provided via the Baraka River system and the Tokar Delta, because the catchment of the rivers does not extend to the ESP but is instead confined to rocks of the Arabian Nubian Shield, which do not weather to produce smectite. Cenozoic volcanic rocks cropping out in the southwestern Arabian Peninsula are a possible source for this smectite (Fig. 1A). A number of small wadis originate from this outcrop (U.S.G.S., 1963) and may have discharged some volcanic weathering products including smectite during this very

intense humid period. Another possible source is the basaltic terrane of the Afar region to the south. Although there are no major rivers draining into the Red Sea or Gulf of Aden today, a palaeodrainage system may have reached the Red Sea during the Eemian. Also, we do not rule out the possibility that extensive rainfall flushed volcanic weathering products to the southern Red Sea and the fine fraction was transported northwards by marine currents.

AHP6 (ca. 170 ka) finds no expression in the fluvial EM3 (Figs. 2, 7g; Suppl. Fig.1). This is probably due to the generally cooler glacial boundary conditions with less intense monsoon activity than during interglacial AHPs (Gallego-Torres et al. 2010). The summer rainfalls in the headwaters of the rivers and wadis probably are inferred to have been too weak to allow the establishment of marked fluvial systems with sediment transport. The diminished smectite concentrations, however, indicate a decline in the dust input from the ESP. The humidity was sufficient to hamper dust uptake, possibly by the moistening

of the soils and the generation of a vegetation cover. A reduction of dust transport during AHP6 is also documented in the eastern Mediterranean Sea (Ehrmann and Schmiedl, 2021).

## 5 Conclusions

- The majority of the terrigenous sediments deposited in the central Red Sea during the last ca. 200 kyr was delivered by
aeolian transport. Several source areas probably contributed to the aeolian sediment influx.

- The most important dust source was the Eastern Sahara Province (ESP) of Sudan and southernmost Egypt. It is characterised by volcanic weathering products responsible for high smectite and Ti contents, high $^{87}Sr/^{86}Sr$ ratios and low $\varepsilon_{Nd}$ values. A second dust source, less significant according to the radiogenic isotope data, is the eastern Arabian

Peninsula and/or Mesopotamia. There, dust sources are rich in palygorskite, illite, chlorite and K contents. Finally,
kaolinite is probably transported as dust from Egypt.

- The strength of the African monsoon is strongly imprinted on our record of dust influx from the ESP via the Tokar Gap
to the central Red Sea. All of the insolation maxima of the last ca. 200 kyr are documented in our records, not only the
strong ones that led to sapropel formation in the eastern Mediterranean Sea.

- Fluvial sediment input as indicated by grain size endmember 3 mainly comes via the Tokar Delta from the Baraka
catchment.

- The intensity of the fluvial influx indicates different strengths of the individual humid periods. Most pronounced was
AHP5, followed by AHPs 7, 4, 3 and 1. No fluvial activity is reported for the glacial AHP6.

**Data availability**

All data will be accessible via the PANGAEA database at the Alfred Wegener Institute for Polar and Marine Research,
Bremerhaven, Germany (https://doi.pangaea.de/10.1594/PANGAEA.XXX). (data uploaded, waiting for doi)

**Author contributions**

WE and GS initiated and designed the research project. WE was in charge of the sedimentological data, PAW of the radiogenic
isotope data. WE and GS wrote the first draft of the paper. All authors contributed to the interpretation and discussion of the
data, and to the writing of the submitted paper.

**Competing interests**

The authors declare that they have no conflict of interests.

**Acknowledgements**

We thank the master and the crew of "RV Meteor", the chief scientist Hjalmar Thiel (Hamburg) and the group leader Christoph
Hemleben (Tübingen) for their efforts during cruise M5/2 in 1987. WE and GS are grateful to Sylvia Haeßner for outstanding
technical assistance in the sedimentological laboratories at the University of Leipzig. Dennis Bunke performed the grain size
analyses and the XRF measurements. Further manifold scientific and technical support came from Stefan Krüger. Thanks also



to Kumpel Ehrmann for assistance and listening. The study is a contribution to the Cluster of Excellence 'CLICCS - Climate, Climatic Change, and Society', and a contribution to the Center for Earth System Research and Sustainability (CEN) of Universität Hamburg". PAW thanks Anya Crocker and Tereza Kunkelova for discussions and Amelia Gale, Yuxi Jin, Matt Cooper and Andy Milton for laboratory assistance.

## Financial support

The German Research Foundation (Deutsche Forschungsgemeinschaft, DFG) financially supported the studies of WE and GS (Eh 89/23-1, Schm 1180/26-1). PAW acknowledges the Royal Society (Challenge Grant CH160073 and Wolfson Merit Award WM140011), NERC (grant NE/X000869/1) and University of Southampton's GCRF strategic development grant 519016. The "Open Access Publishing Fund" of Leipzig University supported by the German Research Foundation within the program "Open Access Publication Funding" covered the publication costs.

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
