# Peer review of "Monsoon-driven changes in aeolian and fluvial sediment input to the central Red Sea recorded throughout the last 200,000 years"

_Climate of the Past, 2023_

## Author Comment (AC1)

**Reviewer 1:** Anonymous

*General comments:*

*This is an excellent paper that addresses the relatively under-studied eolian history of the Red Sea. The subject is of interest to the readership of the paper and will probably draw attention and referenced by future works in the region. It focuses on the central part of the Red Sea around the Tokar Gap. An interesting region that experiences large climatic variations also through winter-summer conditions, and on a larger scale of glacial-interglacial conditions it had shown large variation in the regional climatic conditions. Dust proxies usually present a messy record as there are several sources – specifically in a site that accumulated both eolian and fluvial sediments. Yet, the proxies examined here were discussed appropriately, showing an overall agreement and the text clearly demonstrates this matter. Some differences may arise regarding the interpretation, and I added the fundamental issues below, however, the paper should be published addressing few minor revisions.*

**[Response] Thank you for this very positive assessment of our study and for your suggestions of how to improve the manuscript.**

*Specific comments:*
*The authors argue that a fluvial origin of palygorskite is unlikely, however, there seem to be a correlation between rising proportions of it and EM3, and the occurrences of S events. EM3 is considered a fluvial source and has robustly been shown to be correct in previous works in the Red Sea, even by one of the authors. How then, does this alleged discrepancy settle?*

**[Response] We regard a fluvial origin of palygorskite as unlikely, because no potential source rocks are known in the catchment areas of the small wadis entering the central Red Sea (lines 319–321) from the Arabian and African margins. The rising concentrations of palygorskite during times of enhanced loadings of the fluvial grain size end member EM3 are explained by a dilution effect (lines 408–409). During the humid phases (sapropel events) the main source for clay minerals, the Eastern Saharan Province, was much less active and, thus, the smectite concentrations in KL11 decreased by ca. 20%. Because the concentrations of all clay minerals add up to 100%, the concentrations of illite, chlorite and palygorskite rise as the smectite concentration drops. We have clarified this in the revised version of the manuscript.**

Values of 87Sr/86Sr that are higher than 0.707 accompanied with εNd values lower than -2 are not likely to reflect products of basalt weathering. Perhaps there is another source rock in the area that can provide these values? Could it be related to the palygorskite from the previous comment?

**[Response] Agreed, those are not values reflecting purely basaltic input. What we are trying to say is that the composition of KL11 sediments points to supply from a source where there is a strong imprint of basaltic weathering input. Our data correspond well to the composition of active dust sources in the Eastern Sahara (Fig. 6) where there is a strong imprint of sediments transported down the Blue Nile from the Ethiopian Highlands. We have edited the text to improve our clarity of meaning.**

*Technical comments:*

*Line 178: According to Fig. 3 the smectite minima at MIS5 occurred at 118-119ka and not at 125ka as stated.*

**[Response] The broad Eemian smectite minimum at ca. 129–118 ka is interrupted by a short and small secondary maximum, just in the centre of AHP5 (discussed in lines 423–433). We have changed the number 125 to 128/120 in the revised manuscript.**

*Lines 396-398: The argument, to my understating, is that the alluvial fans channels and wadis stopped providing dust due to their increased hydrological activity, whereas the alluvial plains continued to do so. Thus, the process is wetting rather than desiccation, I advise to use this terminology as the revers is harder to grasp and not chronologically accurate.*

**[Response] We followed your advice and have changed the sentence to: "The peaks in these records correlating to AHP3 and AHP7 occur some 2 kyr earlier than the corresponding smectite minima (Fig. 7f, g) presumably signalling wetting of the alluvial plains after their associated fans, channels and wadis."**

---

## Author Comment (AC2)

**Reviewer 2**: Alberto Reyes

*General comments: This is a solid paper that presents detailed grain-size, clay mineralogy, sXRF, and Sr and Nd isotope data for a core in the Red Sea. The authors persuasively interpret these downhole data in the context of climate-driven changes in eolian and fluvial sediment supply. Connections are drawn to regional humid periods as interpreted from sapropels and other dust flux records. Though I am not a specialist in this field area, the paper was interesting to read and I recommend publication with some minor revision. My only substantive issue is that some additional information should be provided on the core age model and implementation of the grain-size endmember mixing model. I also have suggestions for expanded discussion on a few points.*

**[Response] Thank you for your generally very positive assessment of our study and your useful and constructive comments and suggestions. We deal with them in this rebuttal letter and will revise our manuscript accordingly.**

*Age model: More details on the age model are needed, most importantly estimates of uncertainty. Authors cite Hartman et al 2020 for the upper ~900 cm of core but in quickly looking at this source I was unable to locate any sort of discussion of the age model uncertainty (other than acknowledging that it exists). A quick age-model plot would let interested readers know which parts of the age-scale are based on the new (and essentially undescribed) age model below ~900 cm. Ideally, authors would be able to provide some estimated values for age model uncertainty through various core intervals. Obviously this is critical when assessing dust flux records between sites or comparing the various proxies to sapropel timing.*

**[Response] We can provide additional information about the age model uncertainties. The adopted age model of Hartman et al. (2020) for the core interval above 901 cm is based on radiocarbon ages with average 2σ errors of ±0.46 ka and correlation to the U-Th dated Soreq cave speleothem with average 2σ errors of ±0.84 ka dates (Grant et al., 2012). In the revised manuscript, we provide the tie points for KL11 and the Asian speleothem record, which forms the age model for the core interval below 901 cm core depth. The age uncertainties for this core interval are given by the Asian speleothem U-Th dates with average 2σ errors of ±2.17 ka (Wang et al., 2008).**

*Similarly, please provide more details on EMMA implementation. What were the model parameters for things like convexity error, weighting exponent, etc?*

**[Response] We did not use the programme EMMA (Weltje, 1997) but RECA (Seidel and Hlawitschka, 2015) for computing end members, see line 116. The programme and its algorithm are described in detail by Seidel and Hlawitschka (2015). We used a convexity error of -6, a weighting exponent of 1 and 1500 iterations for a 3-end member model. We have added this information to the method section of our revised manuscript.**

*A few discussion/results items popped out in terms of potentially deserving more discussion or elaboration (though authors may disagree that these are important):*

*- There's a distinct and persistent drop in the loading of EM1 during MIS 6, ~160 ka. Since periods of low EM1 loading are discussed prominently, could authors speculate on this?*

**[Response] The minimum in EM1 (coarse dust) around 165 ka probably correlates with the glacial humid phase AHP6. This AHP was too weak to produce fluvial runoff. However, reduced smectite concentrations indicate that dust transport from the Eastern Saharan Province was reduced, probably by moistening of the soils and the generation of a vegetation cover (lines 435 ff). The minimum in the loading of EM1 additionally may indicate a reduction in wind speed. We have added a statement to our revised manuscript.**

*- The maxima in EM3 seem to consistently predate the timing of sapropels (which authors indicate—in introduction and Fig 7 caption—are associated with AHPs). On the other hand, the EM3 maxima are better aligned in time with the Ti/Al monsoon humidity index in Fig 7d from ODP Site 968. Maybe this is all just age model uncertainty instead of real leads/lags. But could consider elaborating on this in the text.*

**[Response] We assume that the age discrepancies between sapropel events in the eastern Mediterranean Sea and changes in sediment composition of KL11 in the central Red Sea (e.g., EM3, Ti/terr ratios, concentrations smectite) are due to different age models. The age models in the studies of sediment cores from the eastern Mediterranean Sea were mainly based on a correlation of the oxygen isotope records with the LR04 isotope stack (Lisiecki and Raymo, 2005). In contrast, the age model of KL11 (Grant et al., 2012, Hartman et al., 2020) is based on a comparison of the oxygen isotope record with U-Th dated speleothems. The differences in the age models are mentioned in the methods section (lines 101-105). Additionally, we have inserted a statement to the discussion section of our revised manuscript.**

*-I'm not familiar with the Sr and Nd isotope characterization of endmembers in this field area. But my own experience with compiling endmember compositions from the literature and applying those to terrigenous sediments prompts me to ask if it's worth discussing potential grain-size and/or whole-rock biases when it comes to interpretation of the data. This could arise from (1) bulk terrigenous sediment digestion from the KL11 sediments, which would include both clays and coarse silt (and even fine sand), and (2) potential biases from comparing whole rock vs various sediment size fractions for the endmember characterization. Sr isotopes, in particular, may be subject to strong grain size biases. I'm not suggesting this is the case here, but I think it's a point worth addressing.*

**[Response] We agree, these issues, along with the potential for terrigenous signals to be contaminated by marine phases, are not always given adequate consideration. We have added a discussion in section 2.3.**

*Minor comments/suggestions (numbers refer to line numbers):*
*53: Maybe more reasonable to state that AHPs led to ecosystem and hydroclimate change that facilitated open pathways for human dispersal?*

**[Response] Agreed. We have changed the text accordingly.**

*57/58: "maximum" in what?*

**[Response] We have replaced "last interglacial maximum" by "Eemian" in the revised manuscript.**

*87-92: Can you provide some citations to lead interested readers to primary sources for the core (e.g. a cruise report)?*

**[Response] Unfortunately, the cruise report (Nellen et al., 1996) does not contain much basic information on the core. We therefore also include references to the publication by Hemleben et al. (1996) that contains some basic information, and to the PhD thesis by Schmelzer (1998) that contains a rough core description.**

*220: A comparison is made to dust accumulation data from KL09. Perhaps useful to plot these data (Fig. 7?) to facilitate the comparison for readers.*

**[Response] The KL09 data are shown in Supplementary Figure 2. We have inserted an additional reference to this figure in the revised manuscript.**

*231: "...local source for the riverine sediment input..."?*

**[Response] Agreed, has been changed.**

*244: specify the callout to Fig 2B (the time series for the EMs)*

**[Response] Agreed, has been specified.**

*250: Interesting that EM1 and EM2 are generally anti-phase (except for during AHP 5, as noted). Is this worth elaborating on?*

**[Response] Because the loadings of the three endmembers sum up to 1.0, EM1 and EM2 have to be anti-phase if EM3 is more or less constant. No change to the manuscript is needed.**

*271: Very minor point, but "entrained" seems better here than "emitted"...*

**[Response] We have changed "dust is preferentially emitted from" to "dust is preferentially activated from" in the revised manuscript.**